# Targeted Sequencing Approach and Its Clinical Applications for the Molecular Diagnosis of Human Diseases

**DOI:** 10.3390/cells12030493

**Published:** 2023-02-02

**Authors:** Xiao Meng Pei, Martin Ho Yin Yeung, Alex Ngai Nick Wong, Hin Fung Tsang, Allen Chi Shing Yu, Aldrin Kay Yuen Yim, Sze Chuen Cesar Wong

**Affiliations:** 1Department of Applied Biology & Chemical Technology, The Hong Kong Polytechnic University, Hong Kong 999077, China; 2Department of Health Technology and Informatics, The Hong Kong Polytechnic University, Hong Kong 999077, China; 3Department of Clinical Laboratory and Pathology, Hong Kong Adventist Hospital, Hong Kong, China; 4Codex Genetics Limited, Unit 212, 2/F., Building 16W, No. 16 Science Park West Avenue, The Hong Kong Science Park, Hong Kong 852, China

**Keywords:** molecular diagnosis, targeted sequencing, next-generation sequencing, COVID-19 detection, bacteria identification, cancer marker detection

## Abstract

The outbreak of COVID-19 has positively impacted the NGS market recently. Targeted sequencing (TS) has become an important routine technique in both clinical and research settings, with advantages including high confidence and accuracy, a reasonable turnaround time, relatively low cost, and fewer data burdens with the level of bioinformatics or computational demand. Since there are no clear consensus guidelines on the wide range of next-generation sequencing (NGS) platforms and techniques, there is a vital need for researchers and clinicians to develop efficient approaches, especially for the molecular diagnosis of diseases in the emergency of the disease and the global pandemic outbreak of COVID-19. In this review, we aim to summarize different methods of TS, demonstrate parameters for TS assay designs, illustrate different TS panels, discuss their limitations, and present the challenges of TS concerning their clinical application for the molecular diagnosis of human diseases.

## 1. Introduction

Next-generation sequencing (NGS), a new era of technology, is increasingly used in clinical research, cancer biology, and pharmaceutical development with its exquisite resolution, accuracy, and cost-effectiveness. Upon the development of scalability, high throughput, and user-friendly NGS devices, large-scale NGS experiments are now more affordable than before in a reasonable turnaround time [1,2]. This has led to the expanding implementation of NGS from research to the clinical laboratory [1].

There are three types of NGS sequencing, namely whole genome sequencing (WGS), whole exome sequencing (WES), and targeted sequencing (TS). WGS provides the most comprehensive coverage, which is more suitable for novel gene discovery and research applications [3]. WES involves sequencing exomes, which are composed of exons only, and some of the exons are with the coding region for protein translation [4]. Compared to WGS and WES, TS panels focus on a particular cluster of genomic regions and have fewer data burdens with the level of bioinformatics or computational demand [1]. It can simplify data interpretation with excellent coverage depth, facilitating lower cost and faster turnaround times, essential to many industrial and clinical applications where speed and cost are the most important. Prior to the development and the use of the TS panels, it is important to include the additional step of target enrichment for the genomic regions that are of interest and compared to the genomic background. This step is crucial and ensures that the NGS process is specifically designed to sequence the genomic targets efficiently and accurately. Precisely, the process focuses on the amplification of the target gene or sequences of interest, thus allowing high sensitivity and specificity in identifying sequence variations in diseases [5]. The common sequence enrichment processes include the polymerase chain reaction-based amplicon and hybrid capture-based technique, which will be elaborated on further below [6]. Additionally, nowadays, TS has become an important routine technique in both clinical and research settings, with the advantages of high confidence and accuracy and relatively low cost. Since the comparability of different approaches and techniques for mutation profiling still exists, there are many commercial solutions available for researchers or clinicians to choose from in their assay designs.

The sequencing of genomic DNA extracted from normal tissues (germline) and tumours (somatic) are the two most common approaches in research or clinical application for the appropriate treatment decision [7] or making correct prognosis monitoring of cancer patients by comparing mutations through tumour molecular profiling. In addition, throughout the Coronavirus Disease 2019 (COVID-19) pandemic, different target enrichment NGS panels were also developed as the “molecular fingerprint” for viral detection, identification, and characterization of the patient’s sample, with a positive result of COVID-19, surveillance testing, and environmental monitoring [8]. Scientists understand the transmission of disease, tracking strain origin and viral evolution through the full sequence information. However, more efforts are needed to be put into the assay design, including the purpose and scope of the assay, pre-analytic consideration, sequencing, bioinformatics, and interpretation and reporting, for the development of cost-effective approaches for the molecular diagnosis of diseases. Most of the existing pipelines and approaches designed for WGS or WES can be applied in the data analysis of TS. However, due to the requirement for the high depth of coverage in TS, it is critical to make sure only the variant cells with high quality are retained during the data analysis of TS, especially for the data generated from fragmented and poor-quality DNA [1].

In this review, the applications of TS in various clinical and research assays and, also, the important parameters arising from recent studies will be discussed. Moreover, the advantages and limitations of the recent TS panels used to profile clinical samples will be presented in this review as well. This review aims to provide an overview and updates on the use of TS in the field of microbiology and for human diagnostic purposes.

## 2. Targeted Sequencing

Many well-known gene mutations that cause disease pathogenesis such as cancer driver genes have been widely applied in clinical operations. TS panels focus on a selected number of these specific genes for diagnosis, prognosis, treatment monitoring, etc. Therefore, the cost can be reduced, and greater confidence and better insurance reimbursement opportunities will be provided by using TS panels in clinical settings [1].

For profiling different clinical samples with lower tumour contents and DNA quality, such as circulating tumour DNA (ctDNA) and formalin-fixed paraffin-embedded (FFPE), TS provides a greater sequencing depth of coverage (1000× or higher) than the non-NGS-based techniques, such as allele-specific amplification refractory mutation system (ARMS), polymerase chain reaction (PCR), allele-specific PCR (AS-PCR), bead emulsification amplification and magnetics (BEAMing) technology, droplet digital PCR (ddPCR), and Sanger sequencing. This approach is capable of picking out mutations that are only present in a small part of malignant cells and able to detect a variant allele frequency (VAF) as low as 0.1–0.2% in the case of detecting minimal residual disease [1]. In addition, since the mutations that cause truncation or possible mRNA attenuation in any region of the tumour suppressor genes can be considered clinically significant, the technologies mentioned above are impossible to detect the whole regions of tumour-related genes [1].

### 2.1. The History of Sequencing and Discovery of TS

The first DNA sequencing, called Sanger sequencing or original DNA sequencing, was developed by Frederick Sanger et al. in the 1970s [9]. In the following years, Sanger sequencing was continuously improved, such as the replacement of phospho- or tritium-radiolabelling with fluorometric-based detection and improved detection through capillary-based electrophoresis [10]. These improvements made sequencing more efficient and accurate. Next, the pyrosequencing technique was pioneered by Pål Nyrén and colleagues and later licensed to a biotechnology company named 454 Life Sciences. Pyrosequencing can be performed using natural nucleotides (instead of the heavily modified dNTPs used in the chain termination protocols) and observed in real time (instead of requiring lengthy electrophoreses) [11]. These techniques form the backbone and stimulate the development of NGS applications. NGS brings about a revolutionary understanding in basic and clinical research due to the massively parallel analyses, ultra-high-throughput, cost-effectiveness, and accuracy. Although the principles behind NGS and sanger sequencing are similar, NGS can bind millions of DNA pieces by using flowcell and sequencing at the same time, but Sanger sequencing can only sequence one fragment at a time. Currently, there are three major systems of NGS, including (i) the Roche 454 System, the detection of pyrophosphate released during nucleotide incorporation; (ii) AB sequencing by Oligo Ligation Detection (SOLiD); and (iii) the Illumina GA/HiSequ System that is based on Solexa’s Genome Analyzer (GA)—sequencing by synthesis (SBS) [12].

James D. Watson’s genome was the first individual genome sequenced using the Roche/454 NGS platform and was completed in two months by Wheeler et al. and colleagues [13]. WGS investigates the whole genome, including coding, non-coding, and mitochondrial DNA. Another objective of WGS is to discover novel and unknown genomic variants for the target diseases. The first disease-relevant variants were reported in a family with a recessive form of Charcot–Marie–Tooth disease by WGS [14]. Moreover, WGS was widely applied in cancer genome sequencing and provided diagnostic and therapeutic information for cancer patients. WES was developed to capture protein-coding regions of the genome. Compared to WGS and WES, TS focuses on specific genes and coding regions of interest in the genome with greater sequencing depth. The target genes or regions are well known to relate to the pathogenesis of diseases and clinical relevance. For instance, TS panels have been developed for detecting and monitoring cancer-inherited gene mutations and somatic changes and are important for explaining the landscape of genetic mutations that occurs across different cancers. Information on mutations was important to identify novel therapeutic repurposing and make therapeutic decisions. An example is the identification of microsatellite instability in colorectal carcinoma, which can affect the treatment strategy [15]. Additionally, Frampton et al. found clinically feasible mutations in 76% of the 2221 tumours studied; compared with other modern diagnostic tests, including Sanger sequencing, mass spectrometry genotyping, fluorescence in situ hybridization, and immunohistochemistry, the operable detection of drugs has been increased three times [16,17].

### 2.2. Assay Design Consideration for TS

To design the desired TS panels with the customized enrichment probe, the scope and purpose of the assay have to be defined in advance. Briefly, the general NGS workflow is shown in Figure 1.

#### 2.2.1. Genetic Heterogeneity

The genes and genetic variants in an assay were selected by researchers or clinicians based on the target diseases. For instance, genetic heterogeneity is a significant challenge in designing effective strategies for anticancer treatment. It may lead to drug resistance during cancer treatment because of the clonal interactions [18]. Therefore, the highest impact genes, which contribute to the phenotype without associating with multiple conditions, should be chosen. Researchers can use characterized reference materials (RMs), such as Genetic Testing Reference Materials Coordination Program (Get-RM) and the Genome in a Bottle (GIAB) Consortium for assay development, quality control, validation, and proficiency testing, to determine whether it is difficult to sequence the gene or region of interest [19]. Some public databases such as Gen Curation Coalition (GenCC) or ClinGen can help researchers to determine whether the genes of interest are linked with the disease with a scoring matrix. Target variant types such as copy number alteration (CNAs) and gene mutations, including small insertions or deletions (Indels), small nucleotide variants (SNVs), structural variants (SVs), or epigenetic alterations in germline and somatic mutations, can be included in the scope of the assay as well. After defining the region of interest, researchers or clinicians can determine which TS approach to capture the region of interest with consideration of the turnaround time, cost, workflow, and bioinformatic activity.

#### 2.2.2. Pre-Analytical Considerations

Pre-analytical consideration is very important in the assay design for achieving the desired coverage of reading. Moreover, the required specimen types varied by different types of genetic testing. For example, germline testing mostly requires cells in saliva, peripheral blood, or buccal swabs that do not have cancer cells. In contrast, somatic testing is usually taken after a patient has been diagnosed with cancer, and the expected sample types are usually Formalin-Fixed Paraffin-Embedded (FFPE) tissue, fresh-frozen tissue, and cell-free DNA (cfDNA). The minimal quantity and quality of extracted DNA/RNA such as the OD 260/280 ratio, concentration, and fragment size for the downstream procedure are critical to determining suitable approaches and reagent kits to be used during the workflow of TS [20]. The appropriate choice of sequencing platforms, such as the number of samples per sequencing run, the desired read length and level of coverage for the assay, and using paired-end or single reads, are all critical factors that affect the level of coverage and the cost of the assay [20]. Lastly, the strategy of bioinformatics analysis in the designed pipeline, including alignment, variant calling, and tertiary analyses, need to be considered in the assay design as well.

#### 2.2.3. Sequencing Cost-Effectiveness

The evolution of NGS in the past two decades has been phenomenal. The ability to produce large amounts of sequencing data to unravel the human genome sequence at high speed was achieved in 2008 [21]. The costs of the sequencing hardware and consumables are reducing over time; thus, it has been possible to bring NGS into patient care and at the population level [22]. As mentioned, WGS has the most comprehensive coverage but has less depth as compared to WES. However, the cost of WGS compared to WES is much higher by up to five-fold. The average cost of WGS can be up to USD 24,810, whilst WES cost up to USD 5169 [23]. With the WES targeting protein-coding region of genes in the genome only, this allows for reduced costs due to fewer storage requirements and reduced consumables and analysis costs [24]. Additionally, studies have shown that the use of WES or targeted sequencing early in the diagnostic process is able to lower the diagnostic cost and time for patients. Early WES test would benefit patients by reducing the EUR 3025.56 per individual of dispensable examinations [25]. Furthermore, the use of WES in developing countries is favoured. In Jordan, the average cost of a WES for the diagnosis of children with developmental delay using WES is approximately USD 1200. The initial upfront cost may provide early diagnosis, reducing the diagnostic odyssey, and allows genetic counselling to be provided. Early identification and planning will reduce the financial burden on the family and healthcare institute [26]. Additionally, TS is an alternative method that has a much-reduced cost by targeting for single-nucleotide polymorphism (SNP), indels, copy number variations, gene fusions, etc. in disease diagnosis or screening. As the targets are more specific, the runs can achieve higher depths, a shorter running time, lower storage space, and easy interpretation for large-scale implementation. The current cost for a TS panel can be as low as USD 300 [1]. All in all, with the advancement in sequencing technology and the reducing cost, NGS will continue to become more popular as a tool genomic research and for clinical diagnostics.

### 2.3. Method of TS

Amplicon and capture-based approaches are the two main types commonly used in TS. The amplicon enrichment-based approach uses a predesigned specific primer to amplify the regions of interest before the library preparation [1]. This approach is often used in some experiments, in which cost and sample quantity are the factors of consideration in the assay design. On the other hand, in the hybrid capture-based approach, DNA is firstly fragmented and utilizes hybridization oligonucleotide bait attached to relatively long sequence-specific DNA or RNA probes to capture the target region of interest [1,27]. The hybridization can take place either in a solid phase, which involves solid support such as a glass microarray slide for probe attachment, or in the solution where the probes are biotinylated to a magnetic streptavidin bead [27]. Compared with the hybridization-based approach, the amplicon-based approach is cheaper and speedy with a simpler workflow and fewer starting materials. However, there are several limitations. Firstly, the limitation of mismatch tolerance between the primers and the target sequences will increase the risk of amplification failure, because viruses continue to change over time [28]. Secondly, this approach also increases the difficulty in achieving uniform target coverage, especially with a low viral load or poor-quality samples [29]. Some commercial amplicon sequencing platforms try to solve the problem of coverage by using specific primers that can amplify overlapping fragments in a single multiplex PCR (mPCR) reaction [30]. However, it is still challenging to design primers that can enrich certain regions with a high number of repeated sequences. To overcome this issue, the long sequence of bait used in the hybrid capture-based approach allows better specificity in the selected regions. Moreover, since hybrid capture has been demonstrated that fewer PCR duplicates were produced compared with the amplicon-based approach [31], removing the PCR artefacts reduces the possibility of two unique fragments being aligned with the same genome coordinates [1].

## 3. Clinical Applications of TS

### 3.1. SARS-CoV-2 Surveillance and COVID-19 Research

The COVID-19 pandemic emerged in December 2019 and has spread globally affecting millions of individuals worldwide [32,33]. Different approaches for Severe Acute Respiratory Syndrome Coronavirus 2 (SARS-CoV-2), which is the causative agent of this disease have been implemented and developed globally [34,35]. Understanding the genetic epidemiology and evolution of the virus is important, which can rapidly identify the virus for diagnosis and surveillance and prevent the spread of this pandemic disease [36,37]. Considering the high mutation rates of the RNA virus and the virus genome may be mixed with host RNA during sample isolation, this increases the difficulty for primer design and reconstruction of the viral genome. This also compromises the effectiveness of SARS-CoV-2 detection techniques. Therefore, there is a vital need to develop efficient approaches or tests that tolerate mutations and can characterize the viral genome, including genetic variants for rapid detection for diagnosis and surveillance of COVID-19 [29].

Currently, there are different tests for SARS-CoV-2 detection including real-time reverse transcription polymerase chain reaction (RT-PCR), immunoassays, and sequencing-based methods, including shotgun metagenomics sequencing (mNGS) or TS. RT-PCR is widely applied in clinical laboratories with its advantages, such as low cost, fast processing, results acquisition, and easy workflow [27,34,35]. TS for the detection of SARS-CoV-2 has several advantages over other techniques. For instance, TS can detect microorganisms that cannot be targeted by the designed primers and probes since their sequences are too divergent when compared to RT-PCR [27]. It was demonstrated that COVID-Seq identified the presence of SARS-CoV-2 in the samples previously categorized as inconclusive in RT-PCR [38]. Additionally, TS generally has a higher sensitivity and requires fewer starting materials compared with RT-PCR [39]. In addition, TS can able to obtain a partial to complete genome of the virus and provide more information on the genetic diversity, genotype, and virulence of the virus [40]. This allows additional applications such as genomic characterization of the virus, viral surveillance, and variant analysis for viral evolution [39].

mNGS is a high-throughput sequencing approach that genotypes and identifies all microbial communities in samples without any prior knowledge of microbes [41]. The minimum read recommendation of mNGS is 10,000,000 compared with the 500,000 on TS [41]. This approach can discover co-infections, which can get the RNA from the human transcriptome, not only get the RNA from SARS-CoV-2. However, mNGS lacks sensitivity and only can produce the whole genome for the sample with low viral load by using the high depth of sequencing. This issue can be overcome by TS approaches. Specific genes in samples are sequenced with a great sequencing depth (e.g., ultra-deep sequencing depth > 10,000×) by the TS method, which can increase the sensitivity and accuracy and decrease the amount and burden of data requiring analysis and its turnaround time [1,42]. Moreover, TS was also shown to exhibit a higher sensitivity to the target in the presence of high background from the host compared to mNGS [43]. Furthermore, TS is more susceptible to the mutational effect. It can be impacted by a single-nucleotide polymorphism (SNP) or indels located within primer-annealing regions or probe-hybridizing regions, leading to a variation in the optimal annealing temperature and a decrease in the amplification efficiency [42]. As a result, the primers for annealing and hybridization should be constantly updated due to the high evolution rate of SARS-CoV-2. Missing regions with no coverage should require further validation by using mNGS.

Host RNA is included in most of the SARS-CoV2 patient samples, which will be inefficient and costly for direct sequencing [44]. Hence, few commercial kits are available for hybridization-based targeted enrichment of SARS-CoV-2. Currently, the commercial kits in the market included Respiratory Virus Oligo Panel (RVOP) and Respiratory Pathogen ID/AMR Enrichment Panel Kit from Illumina, KAPA SARS-CoV-2 Target Enrichment Panel from Roche and Twist SARS-CoV-2 Research Panel from Twist Bioscience, which is compatible with Illumina sequencing. Respiratory Virus Oligo Panel detects and characterizes around 40 common respiratory viruses including SARS-CoV-2 with human probes as positive controls while Respiratory Pathogen ID/AMR Enrichment Panel Kit targets additionally bacterial and fungal respiratory pathogens with detection of antimicrobial resistance alleles [45,46]. The source of respiratory infections and possible antimicrobial resistance can be identified by using these respiratory panels for targeted sequencing. Compared with the kits from Roche and Twist, RVOP has higher sensitivity on SARS-CoV-2 detection, which can detect as little as two copies of viral spiked into human saliva RNA with the full SARS-CoV-2 genome coverage. A study also supported the use of the Respiratory Virus Oligo Panel as it demonstrated a 100% concordance of SARS-CoV-2 detection between this enrichment tool and RT-PCR [47].

Until now, the PCR amplicon-based approach (e.g., ARTIC) was widely used to sequence SARS-CoV-2 accurately and quickly [36,48]. Multiple kits for amplicon-based sequencing are available on the commercial market, such as Paragon Genomics CleanPlex^®^ SARS-CoV-2 Panel and Qiagen QIAseq SARS-CoV-2 Primer Panel. Illumina COVIDSeq Test was the first NGS test approved by U.S. Food and Drug Administration’s Emergency Use Authorization (EUA) for SARS-CoV-2 sequencing with high sensitivity. A total of 98 regions of the SARS-CoV-2 genome are targeted for amplification and 11 human mRNA targets are included as internal controls [49]. It is proven to have high reproducibility and a high concordance of detection with RT-PCR practices [48]. Moreover, the Ion AmpliSeq™ SARS-CoV-2 Research Panel from Thermo Fisher Scientific targets 237 amplicons for SARS-CoV-2 sequencing with 5 primer pairs specific for human expression control. A study has validated the effectiveness of this assay for sequencing the SARS-CoV-2 genome both from isolates and from nasopharyngeal swabs [49]. Additionally, this assay utilizes the Ion Torrent™ Genexus™ Integrated Sequencer, which automates all the procedures starting from cDNA synthesis to post-run analysis [43]. This automated NGS workflow allows easier adoption with only 5 min of hands-on time. Therefore, it can greatly enhance the reproducibility of results and lab efficiency. Compared to Illumina sequencing, this automated ThermoFisher sequencer is less tedious and more cost-effective with faster results [42]. However, sequencing by detection of hydrogen ions using an Ion Torrent is prone to produce indels in homopolymer regions, especially after long homopolymeric stretches [50]. Therefore, it may produce a less accurate read when compared to Illumina sequencing.

The advantages, limitations, and application in different scenarios of the listed approaches were summarized and shown in Table 1. mNGS is still the gold standard for samples with a high viral load for the discovery of novel pathogens and retrieval of a maximum of information about the virus without any bias. TS through hybrid capture or amplicon is suitable for profiling lower quality and fragmented DNA samples with low cost and high sensitivity.

### 3.2. Bacteria

16S rDNA, the genes encoding the 16S ribosomal RNA (rRNA), is ideal for bacterial identification, since it consists of several conserved and nine variable (V1–V9) regions [51]. The conserved regions are the targets for universal PCR primer design to amplify the genes from a wide variety of microorganisms regardless of the bacterial species; while the variable regions are sequenced for specific genus/species differentiation [52,53].

Conventionally, 16S rDNA sequencing is performed by Sanger sequencing. The protocol could be in-house lab-developed or commercial [54]. One of the commercial protocols available in the market was developed by Applied Biosystems™ (Forest City, CA, USA). MicroSEQ™ 500 is the 16S rDNA Sequencing Kit. The first 500 base pairs of the 16S rDNA gene will be amplified and sequenced, as indicated by the protocol. The DNA sequence of the microorganism is analysed on the Applied Biosystems™ (Forest City, CA, USA) Genetic Analyzer using capillary electrophoresis and aligned to the MicroSEQ™ 16S rDNA 500 Library using MicroSEQ™ ID Analysis Software version 3.1, to generate reports of the bacterial identity up to species level [55].

The technology of NGS is evolving and there are multiple commercially available sequencing panels targeting the 16S rRNA, and often, along with the Internal Transcribed Spacers (ITS) rRNA genes [56]. ITS is the non-functional RNA segment between structural rRNA segments in the precursor transcript, which serves as the marker for fungi identification [57,58]. One of the most commonly used NGS platforms for 16S rDNA sequencing is the Illumina MiSeq System. The protocol covers variable regions V3 and V4 of the rRNA gene.

The workflow starts with an amplicon PCR to amplify the selected 16S rDNA region, followed by a post-PCR clean-up. The PCR products then undergo an index PCR to attach dual indices and Illumina sequencing adapters using the Nextera XT Index Kit, and the libraries are subjected to perform a second clean-up. After that, the qualified amplified libraries are normalized and pooled, then ready to be loaded into the Illumina MiSeq System [58]. The generated data are finally analysed by using Illumina softwares. For example, 16S Metagenomics version 1.1.1 is an app to interpret 16S rRNA targeted amplicon reads to classify bacterial taxonomy using the GreenGenes taxonomic database devised by Illumina, whereas ITS Metagenomics analyses fungal rRNA targeted amplicon reads using the UNITE taxonomic database [56]. Likewise, other manufacturers also provided library preparation kits compatible with the Illumina Sequencing System with improved protocols. For instance, QIAGEN QIAseq 16S/ITS Index Kits include 3 pools of primers with 6 amplicons to cover the whole 16S rDNA plus the ITS (Pool 1 covers V1V2, V4V5, and ITS; pool 2 covers V2V3 and V5V7; and pool 3 covers V3V4 and V7V9). In addition, it uses phased primers, which add 0–11 additional bases to the 5′ end of the 16S rDNA or ITS primers. It shifts the nucleotide balance, increases base diversity, improves base-calling quality, and eliminates the need to spike in PhiX [59]. On the other hand, Thermo Fisher Scientific Ion 16S™ Metagenomics Solution is an alternative 16S sequencing panel employing Ion Torrent sequencing systems. The Ion 16S™ Metagenomics Kit includes two pools of primers to amplify seven hypervariable regions (V2, V3, V4, V6, V7, V8, and V9) of the 16S rRNA gene, improving bacterial identification by more comprehensive sequencing of the 16S rRNA gene [60].

#### 3.2.1. Usefulness and Clinical Benefits of Targeted 16S rRNA Gene Sequencing

16S rDNA sequencing is a useful method for bacterial identification and complementary to culture-based methods, such as phenotypic biochemical tests, and Matrix-Assisted Laser Desorption/Ionization-Time-of-Flight (MALDI-TOF) mass spectrometry (MS) [61]. Traditional phenotypic identification has several loopholes that could be plugged by 16S rDNA sequencing. First, the phenotypic profile of the unknown isolate might not be consistent with the typical known pattern. Second, these approaches could not be applied to microorganisms, which cannot be cultured in the laboratory. Third, fastidious or slow-growing microbes (e.g., *Mycobacterium* species and anaerobes) might require extra time and resources for identification. Taken together, sequencing is the pivotal method for difficult bacterial isolates with atypical phenotypic profiles, rare bacteria, uncultivable bacteria, or slow-growing bacteria [62]. The advancement of 16S rDNA NGS panels adds merits by offering direct sequencing from biological specimens without pure cultures, and the potential to study the microbiome in complex specimen matrices such as faeces [56]. The profile of the relative abundance of microbes in a biological sample could be generated in a single analysis, making comparative studies of microbial communities possible (metagenomics). Metagenomics is the future direction in microbiological studies to identify all microorganisms (bacteria, viruses, and fungi) in the specimens [63,64]. The comprehensive sequencing of multiple variable regions of the 16S rRNA gene further improves identification accuracy, which helps in outbreak investigation, infection control, and epidemiology studies [60,64,65]. Moreover, parallel sequencing of multiple samples in a single run is possible, which saves manpower and time [54,58]. Compared to WGS of the bacterial genome, targeted 16S rRNA sequencing simplifies the workflow, minimizes the huge amount of data generated, eliminates the complicated bioinformatics analysis, and reduces the turnaround time. Therefore, the ease to implement 16S targeted sequencing in a clinical microbiology laboratory is better than WGS [63,64,65].

#### 3.2.2. Limitations and Challenges of Targeted 16S rRNA Gene Sequencing

Although 16S rRNA gene sequencing is a powerful approach to understanding the relationship between disease and the microbial community, there are some limitations. First and foremost, it has a discrimination problem at the species level in some genera [62,65,66] shown in Table 2. The genus *Bacillus* is a case in point. The two species, *B.globisporus* and *B. psychrophilus* exhibit > 99.5% similarity in their 16S rDNA sequences, while they only share 23% to 50% relatedness at the DNA level, as shown in reciprocal hybridization reactions [67]. This implies that relying on 16S rDNA sequencing solely might not differentiate species confidently, even though the two species are of low relatedness.

The application of NGS has additional issues to be considered compared to 16S rDNA Sanger sequencing. Every step of the procedures could introduce bias to the sequence libraries, hindering correct data analysis [64]. First, specimen handling, storage, and transport can alter the microbiota profile and the relative 16S rRNA abundance. Second, the DNA extraction protocol should ensure effective lysis and recovery of microbes. Third, contaminating 16S rDNA from the environment, reagent or consumables could disrupt sequencing analysis. Fourth, the major concern is involving the intrinsic error rate and chimaera generation of sequencing, which will create artefacts during the PCR amplification and result in incorrect new species identification and wrong classification [63]. The absolute quantification of bacteria is not possible due to the nature of sequencing. For instance, differential GC content and primer affinity dictate PCR amplification. Furthermore, 16S rRNA gene sequencing can only reveal a microbial profile at a given time, hence determination of cause-and-effect relationships is impossible.

### 3.3. Human

Unlike primary human specimens such as nasopharyngeal swabs, sputum, and stool, cultured colonies derived from microbiology specimens permit extraction of nucleic acid multiple times, if the sample quality following the first attempt is deemed suboptimal. Ethylenediaminetetraacetic acid disodium salt dihydrate (EDTA)-preserved blood or bone marrow is often used for molecular diagnosis of inherited disorders. The detection of somatic mutations can be challenging, given that fixation of the required specific tissues with FFPE induces inevitable damage to nucleic acid. By contrast, cell-free DNA (cfDNA) enables non-invasive prenatal testing (NIPT) and oncology testing for residual mutant transcripts.

Molecular profiling of tumours by NGS sheds light on precision medicine. Therefore, TS gene panels are developed and designed to study different cancer types using tissue and liquid biopsy samples. Considering the cost of WGS/WES, TS panels are preferred in both preclinical and clinical settings due to their inherent design to exclude less-relevant genes for easier data interpretation and improved detection sensitivity.

#### 3.3.1. FFPE

The highly fragmented DNA of FFPE samples present challenges to molecular diagnosis [68]. Depending on the extraction method and sample type, the length of fragmented DNA can range between 150 bp and 350 bp, and the proportion of double-stranded DNA (dsDNA) may be lower than 50% [68]. To meet the sample requirements of FFPE NGS, the quantity and integrity should be examined by fluorescent dye-based methods, such as the dsDNA mode of an Invitrogen Qubit fluorometer or capillary electrophoresis-based methods such as solutions provided by Agilent 2100 Bioanalyser and TapeStation expressed in DIN, respectively. In addition, FFPE QC kits, such as the Illumina FFPE QC and DNA restoration kits, are options to evaluate the sample quality and repair degraded FFPE DNA samples by using a real-time PCR assay, such that the input DNA is eligible for the preparation of NGS libraries.

To overcome the difficulties of low yield and quality of sample DNA, many sequencing approaches rely on extensive PCR amplification. The process leads to an accumulation of sequence artefacts [69], which are based on changes introduced to the original sequence after extraction and are observed more frequently with DNA derived from FFPE samples than their fresh frozen tissue counterparts [69]. C > T transition is an example of known sequence artefacts, a result of the deamination of C, and is often observed with C preceding G [70]. These sequence artefacts can be mistaken as false-positive point mutations [71]. Provided that naturally occurring deamination of C to T is corrected by uracil-DNA glycosylase (UDG), the incorporation of such a repair process during DNA extraction from FFPE tissues can alleviate undesirable C > T artefacts [68]. The discordance of single nucleotide variants detection and indels detection between fresh frozen tissue and FFPE tissue is 1.2% and 1.75%, respectively. Activities during formalin fixation such as deamination contribute to most of the discordance and are suggested to be attenuated by adopting a higher coverage threshold [72].

To date, many commercially available predesigned panels for FFPE-derived DNA sequencing are compatible with various high-throughput sequencing platforms such as Illumina and Ion torrent. AmpliSeq Cancer Hotspot Panel v2 and Archer^®^ VariantPlex^®^ Solid Tumor Kit are predesigned panels with a target of 50 and 67 common cancer-related genes, respectively. AmpliSeq *BRCA1* and *BRCA2* Panel is another predesigned panel that can be used for DNA extracted from FFPE tissue. Although *BRCA1* and *BRCA2* are the only targets, both of which are related to hereditary breast and ovarian cancers, the panel can detect all exons and 10–20 bases at exon–intron junctions. AmpliSeq™ for Illumina Focus Panel enables concurrent sequencing of both DNA and RNA. The panel incorporates 52 genes relevant to solid tumours. In addition to commercially predesigned panels, customized panels are also designed. Lippert. et al. developed a sensitive, cost-effective approach along with the amplicon-based TS and designed a panel for precise and early detection of high-risk HPV by sequencing at both RNA and DNA levels. Intriguingly, a panel design to detect 5610 amplicons from a selection of 156 genes is shown to exhibit a similar efficiency in mapping variants as whole genome sequencing and whole-exome sequencing but at a lower cost with fewer variants of uncertain significance [73].

#### 3.3.2. cfDNA and Circulating Tumour DNA (ctDNA)

cfDNAs are short fragments of DNA (~160 bp) released into the bloodstream in a small quantity following cell death [74]. Tumour DNA (ctDNA) only accounts for a subtle fraction of cfDNA; hence, the accurate detection of rare variants (<1%) from low ctDNA input is daunting [75]. In maternal blood, 3–13% of total cell-free DNA is of foetal origin (cffDNA) [76]. For example, Down syndrome (DS), a result of trisomy of chromosome 21 can be detected readily by targeted sequencing panels without presenting risks of miscarriage associated with conventional prenatal testing including chorionic villus sampling (CVS) or amniocentesis. Not only does a standard prenatal aneuploidy screening manifest a very high sensitivity (99%) and specificity (99.5%) [77], but the test can also be performed as early as 10 weeks of gestation.

Haemolytic disease of the foetus and newborn (HDFN) refers to the placental transfer of maternal allospecific IgG antibodies, through which foetal red blood cells are destroyed [78]. With targeted panel sequencing and NGS, an intervention can be prescribed accordingly in anticipation of such a haemolytic situation. For instance, the administration of anti-RhD antibodies to an RhD-negative mother can prevent the development of HDFN in her RhD-positive foetus [79]. Moreover, sex determination and identification of monogenic disorders can be achieved with next-generation sequencing.

It is also noteworthy that, however, maternal physiological conditions such as post-transfusion or a history of organ transplant may result in false positivity. While the sensitivity of NGS is high enough for diagnostic purposes, pathogenic findings from low coverage or low read depth results should be interpreted with caution and, thus, may not constitute a confident medical report. The yield of cfDNA available is usually small and insufficient to run another replicate.

cfDNA and ctDNA are essential in tracking disease progression in cancer patients. Contrary to FFPE tissue samples, cfDNA can be sampled in a non-invasive manner over time In the context of non-small cell lung cancer cytological samples, the SiRe NGS gene panel detects cfDNA mutation of *EGFR* (MIM:#131550) and *KRAS* (MIM:#190070), *NRAS* (MIM:#164790), *BRAF* (MIM:#164757), *c-KIT* (MIM:#164920), and *PDGFR* (MIM:#173410). Treatment options including cetuximab, erlotinib/Gefitinib, or crizotinib can then be determined accordingly. For cfDNA, cancer profiling can be useful compared with the traditional method with the scarce sample amount, which could be difficult to represent complete driver mutational gene detection [80].

For a healthy individual, possible driver mutation can be identified by using cfDNA following liquid biopsy. Using Oncomine^TM^ cfDNA assay, 7 cases of relevant gene mutation (*TP53* and cancer-related) are found among the 114 healthy, mammogram-confirmed breast cancer-negative donors. This indicates that cfDNA is a potential tool for screening individuals who are at risk of cancer [81].

Cancer detection by cfDNA analyses, unfortunately, indicates neither the tumour location nor the type of cells involved. Epigenetic alterations are known to be tissue-specific during earlier stages of cancer, which can also differentiate recurrent mutations of normal from tumour cfDNA [82], thereby supporting the radiographic diagnosis. A study using a commercial sequencing panel was done [83]. Digital Sequencing^TM^ is a comprehensive sequencing panel of over 50 cancer-related genes. Its sensitivity for cell-free tumour DNA in blood and tissue samples is found to be 85% and 80.7%, respectively [83]. However, the sensitivity in patients with advanced- and early-stage NSCLC is 80% and 54.6%, respectively [84]. Such phenomenon can be attributed, at least in part, to the fact that ctDNA only represents a small fraction of total circulating DNA.

Trusight Oncology 500 ctDNA, a sequencing panel manufactured by Illumina, is designed to detect low-frequency somatic mutations, tumour mutational burden (TMB), and microsatellite instability (MSI) using circulating ctDNA by a hybrid capture approach to enrich 523 clinically relevant genes [85]. According to a study by the American Association for Cancer Research using 5 healthy individuals’ blood with a ctDNA quantity as low as 20 ng, and procured standards, the panel exhibits > 99% sensitivity for SNVs and >98% for indels [85]. Roche’s AVENIO ctDNA analysis kit requires a long preparation time of up to 5 days but is capable of detecting an expanded panel of 77 genes, a targeted panel of 17 genes, and a surveillance panel of 197 genes. The total panel size is 192 kb and allows the detection of SNPs, Indels, and copy number variations. Although the aforementioned panels manifest similar sensitivity and specificity [75], the TSO500 panel requires a higher cfDNA input (30 ng) and a higher number of sequencing reads because of its larger panels (500 genes), and is thereby less appropriate in the clinical setting [74]. Both panels are primarily used for biomarker identification, MSI, and TMB estimation, but not in vitro diagnosis (IVD). Emerging evidence suggests that urine is an alternative to blood as a source of cfDNA for the detection of bladder cancer, should the circulating level of cfDNA be too low for molecular diagnosis [86,87].

#### 3.3.3. TS Approaches for Gene Fusion

Fusion genes are heterozygous genes produced by the juxtaposition of two previously independent genes, followed by structural rearrangements, such as inversions, deletions, translocations and duplications between different chromosomes or within the same chromosome [88]. More than 10,000 gene fusions have been identified in human cancers and many of which are strongly driving changes [88]. Additionally, the approval and development of new drugs targeting rare gene fusions require in-depth molecular characterization of cancer specimens for providing patients with the ideal treatment options. Although different TS or NGS panels for gene fusion analysis have been implemented in the routine procedure of some laboratories, the test for gene fusion is still facing many challenges. The hybrid capture-based and amplicon-based TS approaches can be applied to the DNA and RNA analysis level for gene fusion detection. The DNA-based techniques allow the characterization of the precise gene fusion breakpoints together with other genetic changes, while RNA-based approaches are to identify the expressed fusion genes only and can quantify fusion transcripts, and discriminate splicing isoforms [88]. The pros and cons of using different TS approaches for gene fusion analysis are also listed in Table 3.

Various customized and commercial panels of amplicon-based gene fusion analysis, powered by mPCR to amplify the fusion variants of interest through the use of specific primers flanking exon-exon fusion combinations, have already been validated. However, RNA-based fusion panels also include testing for imbalances in expression between 5′ to 3′ regions of the target gene, so even if the fusion partner is unknown and not included in the panel, the presence of rearrangements can be identified [88]. The Oncomine Solid Tumor Fusion Transcript kit, a classical mPCR approach from Thermo Fisher Scientific is available to analyse approximately 70 gene fusions involving *ROS1*, *ALK*, and *NTRK1*. With a lower RNA sample input (10 ng), a success rate of 99% can be attained [89]. Other RNA amplicon-based approaches and anchored mPCR allow the analysis of unknown and known variants of fusion, since only one of the fusion partners needs to be targeted [88]. Hindi et al. used a tailormade Archer Anchored Multiplex PCR panel to analyse 84 prospective and 72 retrospective FFPE cases with 100% specificity, sensitivity, and reproducibility [90]. However, common PCR-related drawbacks, including non-specific primer binding, allele dropout, and primers dimerization, also apply to anchor mPCR and classical mPCR.

As for gene fusion analysis using hybrid capture approaches, DNA hybrid capture panels are more common than their RNA counterparts. FoundationOne CDx—Foundation Medicine (Roche) and the Memorial Sloan Kettering (MSK) Integrated Mutation Profiling of Actionable Cancer Targets (IMPACT) are FDA-approved DNA hybrid capture panels designed to analyse copy number changes, mutations, and structural rearrangements in 324 and 468 cancer-related genes, respectively, in conjunction with assessment of TMB and MSI [88]. On the other hand, the commercial RNA hybrid capture panels are available from Agilent and Illumina, which are SureSelect all-in One Solid tumour and Trusight RNA fusion panels, respectively. Relative to RNA, a higher stability of DNA can permit a more comprehensive molecular characterization of tumours. However, the sensitivity of a DNA-based panel can be diminished in the presence of fusion breakpoints in long intron regions, which the hybridization capture probes cannot recognize [91]. Davies et al. also demonstrated that RNA and DNA breakpoints are not matched, so it may not be able to predict the gene fusion expression at the DNA level [92].

#### 3.3.4. TS Applications in Rare Disease

Currently, the main complication for establishing TS applications is to link phenotypically similar patients diagnosed with rare diseases to the gene mutation or molecular cause of the disease with statistical analysis and validation. There were some algorithms or platforms [93,94,95,96] developed to discover patients with common phenotypes with specific gene disruption. Nevertheless, those computational tools were not able to connect and unify the different databases for the identification of cases with similar phenotypic and genotypic profiles through standardized applications and procedures. To solve this, Matchmaker Exchange (MME) [97] was developed in 2015 and provided a systemic approach to the analysis of genotypes and rare phenotypes from different databases through a federated network. MME was connected to eight genomic matchmaking databases, including DECIPHER [98], GeneMatcher [99], PhenomeCentral [96], MyGene2 [100], seqr [101], Initiative on Rare and Undiagnosed Disease [102], PatientMatcher [103], and RD-Connect Genome-Phenome Analysis Platform [104]. Until January 2023, through those eight databases, MME has collected more than 30,000 unique genes and more than 200,000 cases [105]. Using those genotypes and phenotypes information, researchers were allowed to use MME to perform standardized matchmaking analysis to discover novel genes-disease association, and more than 20 novel genes were discovered to associate with different types of rare diseases [106]. For instance, gene *ZSWIM6* [107], *WASF1* [108], *ZMIZ1* [109], *VARS* [110], and *DLL1* [111] were found to be associated with various types of neurodevelopment disorders, while gene *TRIT1* [112] and *WDR26* [113] was found to be related to rare genetic disorders characterized by developmental delay and intellectual disability. In short, it illustrated MME can be used as a tool to screen genes of interest for TS panel design for neurodevelopment and other rare genetic disease detection.

Some rare genetic mutations lead to dysmorphic and unique facial features appeared in patient photographs. Recently, there was more research investigating the relationship between facial phenotype and gene-causing rare diseases with the aid of deep learning. Gurovich et al. developed DeepGestalt, a DCNN-based deep learning algorithm, to classify more than 200 syndromes using typical facial features of patient photographs and further predicting syndromic-specific genetic mutation [114]. Furthermore, Hsieh et al. reported that GestaltMatcher, an advanced DCNN-based deep learning algorithm, diagnosed more than 1000 syndromes using patient photographs [115]. This indicated deep learning can support screening patients with rare disorders and predict specific genes mutation for facilitating future TS panel design.

## 4. Challenging in Mutation Identification Genes/Diseases for Target Panels

Other than the limitations and challenges found in various types of primary specimens, various pitfalls also exist in targeted sequencing panels.

### 4.1. Inborn Error of Metabolism NGS

Compared to Sanger sequencing and other hotspot screening techniques, one of the technical breakthroughs of massively parallel sequencing is the ability to investigate multiple genes. Diagnosis of early-onset disorders by NGS in neonatal or paediatric patients may provide disease insight before results of clinical biochemistry become available. Being one of the most severe complications of an inborn error of metabolism (IEM), neonatal hyperammonemia is lethal in infants with inherited urea cycle disorders such as citrullinemia. Clinical manifestations develop after feeding, through which substrates for enzymes or associated proteins (such as transporter) are provided for the generation of a spectrum of metabolites. Conventional diagnosis involves biochemistry analyses of blood gas composition and plasma amino acid profiles by tandem mass spectrometry [116].

Most IEMs are inherited in a monogenic manner, but the symptoms of different IEMs can be similar. For example, hyperammonaemia can be associated with elevated plasma/urine citrulline (>1000 µmol/L), elevated orotic acid in urine, and reduced/absence of argininosuccinic acid in plasma/urine, all of which are indicative of citrullinemia [117,118]. Likewise, patients with argininosuccinic acidemia or pyruvate carboxylase deficiency can also present similar blood biochemistry [118]. Clinicians should exercise caution during the interpretation of such results. Transient hyperammonemia can be found in a premature newborn without IEM [119] but patients with Type II citrullinemia or citrin deficiency (MIM:#603471) can also be asymptomatic until adulthood.

Clinically heterogeneous metabolic disorders can be diagnosed by targeted sequencing panels for IEMs such as the AmpliSeq for Illumina Inborn Errors of Metabolism Research Panel or CleanPlex series of IEM panels. Using DNA extracted from the heel prick dry blood spot on the 2-day-old infant, 594 IEM-associated genes can be sequenced by the described Illumina IEM panel [119]. Before the heel prick, NGS speculated genes under the guidance of biochemical data were investigated by exon-by-exon Sanger sequencing. Despite that conventional molecular diagnosis may not lead to the initiation of treatment in a prompt manner, NGS also has its limitations. Using the aforesaid example of type II citrullinemia, during the analysis of its causative gene, *SLC25A13* (MIM:#603859), low coverage of reads at a certain genomic region including exon 1 requires “gap-filling” by Sanger sequencing. Additionally, the two highly prevalent variants in the East Asian population, namely mutation [I] (c.851_854del) and mutation [III] (c.1638_1660dup) require Sanger sequencing confirmation as suggested by ACMG [120] because patients with the inherited defect can be asymptomatic.

### 4.2. Mitochondrial DNA NGS

Mitochondrial DNA (mtDNA) can either exist in identical copies (homoplasmic) or as a population comprising different variants (heteroplasmic). According to the guidelines of the American College of Medical Genetics and Genomics/Association for Molecular Pathology (ACMG/AMP in 2020, there is an absence of a well-defined threshold of disease-causing heteroplasmy [121]. To distinguish from nuclear mitochondrial DNA-like sequences (numtDNAs), which are sequence homologs of mtDNA, the amplicon-based method is usually preferred in mtDNA NGS for selective amplification of mtDNA due to primer specificity [122]. Common protocols to sequence the 16kb mtDNA genome include the Illumina iSeq 100, MiniSeq, and MiSeq systems [123]. Although the variant heteroplasmy calling is more accurate with fewer amplicons (up to 5.5% heteroplasmy discrepancy between the nine-mtDNA-overlapping-amplicon protocol and the two-mtDNA-amplicon protocol) [124], the pitfall of the initial mtDNA PCR become more obvious if either one of the amplicons generated in a suboptimal manner.

Reproducibility of the mtDNA PCR efficiency is greatly influenced by the mtDNA large rearrangement, which is commonly found in mitochondrial encephalomyopathy, lactic acidosis, and stroke-like episodes (MELAS) syndrome, leading to unavailable PCR bias or allele dropout [125]. Heteroplasmic assessment of other mtDNA variants may therefore be underestimated, hence mistaken as tolerated pathogenic variants with a low heteroplasmic load. Although urine provides an alternate, non-invasive source of mtDNA, the inherent salt content and other impurities are also sources of PCR inhibitors.

Despite the limitations of mtDNA NGS, there are several advantages over conventional Sanger sequencing and WGS. An example of such is to identify mtDNA variants and their respective levels in a single assay. As an accepted standard, Sanger sequencing is highly accurate for variant identification, but it is not cost-effective to sequence the entire mtDNA genome or hotspot screening (e.g., m.3243A > G for MELAS syndrome). The ACMG/AMP has revealed that the reliable heteroplasmy level assessment of Sanger sequencing is only down to 30–50%, whereas mtDNA NGS can achieve a level of 1.5% [121]. Following Sanger sequencing, additional assays such as digital droplet PCR or qPCR may be required to confirm the level of heteroplasmy. In contrast, bioinformatics analyses can map the sequencing reads obtained from mtDNA to either the nuclear DNA genome (hg19) or the revised Cambridge Reference Sequence of mtDNA genome (rCRS/NM_012920.1), but factors including insufficient read depth of mtDNA and falsely mapped numtDNAs may contribute to the inaccurate quantification of variants (i.e., a combined count of both mtDNA and numtDNAs). Exposed to reactive oxygen species, mtDNA is highly vulnerable to mutations, which exacerbates the challenges in accurate mapping and differentiation between true mtDNA and numtDNAs, assuming there is no introduction of sequencing artefacts during fragmentation and sequencing by synthesis (SBS). In contrast, using the iSeq system as an example, the amplicon-based mtDNA can achieve 10,000× of reading depth which facilitates bioinformatics analysis and provides a rare variant with a low heteroplasmy level and a sufficient read count.

### 4.3. Polycystic Kidney Disease NGS(PKD1/PKD2)

Mutation in *PKD1* accounts for 75–85% of the reported cases of autosomal dominant polycystic kidney disease (ADPKD), which is the most common inherited cystic kidney disease with an incidence of 1 in 400 [126]. Mutation analyses of *PKD1*, a gene that is 54 kb-long with 46 exons in which its first 32 exons share 6 pseudogenes (*PKD1P1-P6*) with a ~97% sequence homology [127], are daunting. Moreover, variants of *PKD2* (MIM:#613095) share a high allelic heterogeneity with *PKD1* and contribute to ~15% of ADPKD [128]. As such, amplicon-based NGS utilizing long-range PCR is undeniably more appropriate than the conventional exon-by-exon-based Sanger sequencing for the sequencing of *PKD1*.

For monogenic disorders such as ADPKD, amplicon-based, targeted sequencing panels for both *PKD1* and *PKD2* using the multiplex-dual-index approach are available [129]. Fragments of *PKD1* and *PKD2* are usually amplified by long-range PCR, resulting in 4–5 long amplicons covering the entire gene. Fragmented *PKD1* and *PKD2* amplicons are then given separate barcoded dual-indexes such that both genes can be analysed as a single primary panel before considering other potential ADPKD-related genes. Compared to WES, TS, particularly for the first 32 exons of *PKD1*, exhibits an increased coverage depth and genotype quality [130]. However, index hopping is a drawback of the multiplex dual index approach [131]. Errors in index alignment are more likely to occur when there are multiple indexes in a single NGS run and in conjunction with the use of a 2-plex dual index. Index hopping occurs when wrongly ligated i5 or i7 index to a library during the exclusion amplification leading to false mapping during bioinformatics analysis. In the context of *PKD1/PKD2*, some PKD2 reads may be mapped to PKD1 and generate false-positive variants (Figure 2) [129].

However, index hopping is a drawback of the multiplex-dual-index approach concerned (Figure 2). With more indexes adopted in a single NGS run, especially if a 2-plex dual index is used for a single sample, the recombination of index misassignment is more prone to occur. For a standard dual-index combinatorial 96-well plate, indexes repeat across the rows and down the columns in which all 12 wells in the A-row share the same i5 index but different i7 index (*n* = 12), and vice versa (Figure 3A) [131]. In addition, having other concern of inter-batch index carryover, indexes adopted in an NGS run are usually not overlapping with either the i5 and i7 indexes used in the previous run, hence the indexes adopted in the same batch are commonly distributed in a rectangular manner across the plate, or in other words, intra-batch indexes are adjacent to each other (Figure 3B) [131]. Index hopping occurs when wrongly ligated i5 or i7 index to a library during the exclusion amplification leading to problematic mapping during the bioinformatics analysis. In the case of index hopping in a *PKD1/PKD2* NGS run, a portion of *PKD2* reads may be mapped with *PKD1* and generate false-positive heterozygous variants (Figure 3C) [130].

The dataset generated is much smaller than WES due to the nature of targeted sequencing panels. The smaller throughput system, such as iSeq 100, is more suitable for the purpose, and yet iSeq 100 utilizes the patterned flow cell whose incident rate of index hopping is nearly 10 times the non-patterned flow cell used in MiSeq system. In response to the emerging concerns over index hopping, Illumina has released a whitepaper recommending the use of unique dual indexes to prevent any possible index hopping [132].

## 5. Conclusions

TS renders new approaches practical in various perspectives of clinical diagnosis, and a wide range of predesigned panels developed by different manufacturers are available on the market to cater to the needs of corresponding diagnostic proposes. In the facet of human genetic testing, TS provides the potential to simultaneously sequence multiple designated variants/mutations and can be applied in numerous sample matrices, ranging from the conventional whole blood or FFPE to cfDNA in plasma and urine. In addition, TS plays a role in pathogen detection, identification, and resistance profiling by parallel sequencing. TS has the inherited strength of minimizing manpower, time, and the sequencing data analysis by merely probing into the interesting targets, thus being economical and cost-effective. Nevertheless, it is subjected to artefacts, errors, and biases introduced in the procedures. Additionally, TS is incapable of detecting novel variants aside from the designed targets, because it is confined to a panel of targeted genes. Although the TS panel has its virtues in its implementation in clinical diagnostic settings, the majority of the panels are for research use only and are not intended for diagnostic use at present. Instead of kit verification, further in-house evaluations with the known positive cases confirmed by standard methods and commercial positive control materials are required to claim that these panels are suitable for diagnostic use, and they should be used complementary to other molecular techniques, including qPCR, conventional Sanger sequencing, or digital PCR.

## Figures and Tables

**Figure 1 cells-12-00493-f001:**
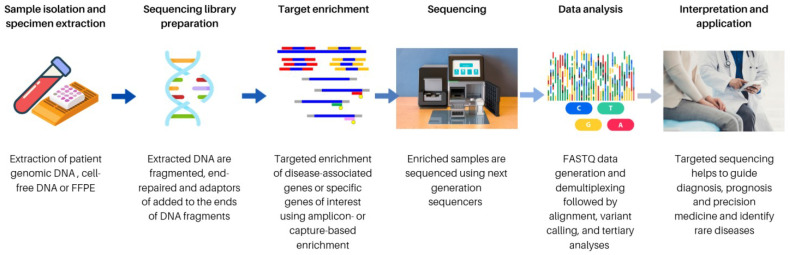
General workflow of a NGS experiment [6]. Firstly, nucleic acid extraction and isolation is performed on specimen types, including blood and FFPE. Next, the library preparation is performed, and it includes fragmenting, performing end repairs, and the addition of adaptors to the nucleic acid fragments. Target enrichment is highly important when performing targeted sequencing to enrich the genes of interest. For sequencing, the right choice of platform, such as the number of samples per sequencing run, desired read length, and level of coverage for the assay, and using paired-end or single reads also are critical factors to affect the level of coverage, and the cost of the assay is taken into consideration prior sequencing. Lastly, the methods used in the bioinformatics analysis in the designed pipeline, including alignment, variant calling, and tertiary analyses, will be conducted for interpretation and application.

**Figure 2 cells-12-00493-f002:**
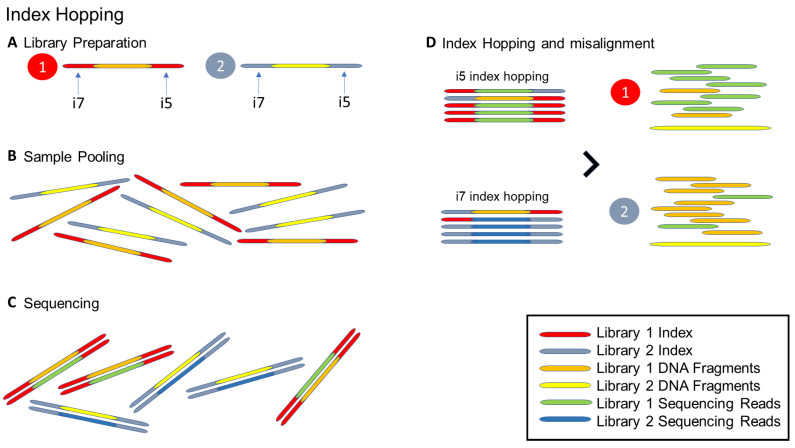
Illustration of (**A**–**C**) index hopping and (**D**) the resultant misalignment for two different samples (libraries). (**A**) During the library preparation, the unique Illumina i5 and i7 indexes would prepare and attach to individual sample DNA fragments. (**B**) After each sample had been uniquely indexed, the two samples can be mixed and ready for sequencing. (**C**) During sequencing, the demultiplexing algorithm would read the sample i5/i7 index and the indexes. Once all the indexes were finished reading, the sample read would be available for the downstream data analysis. (**D**) Through index hopping processing, some i5 or i7 indexes could be wound up across samples and affect the reading. Such a misalignment of sample reading would interfere with the actual interpretation of the results in a downstream bioinformatic data analysis.

**Figure 3 cells-12-00493-f003:**
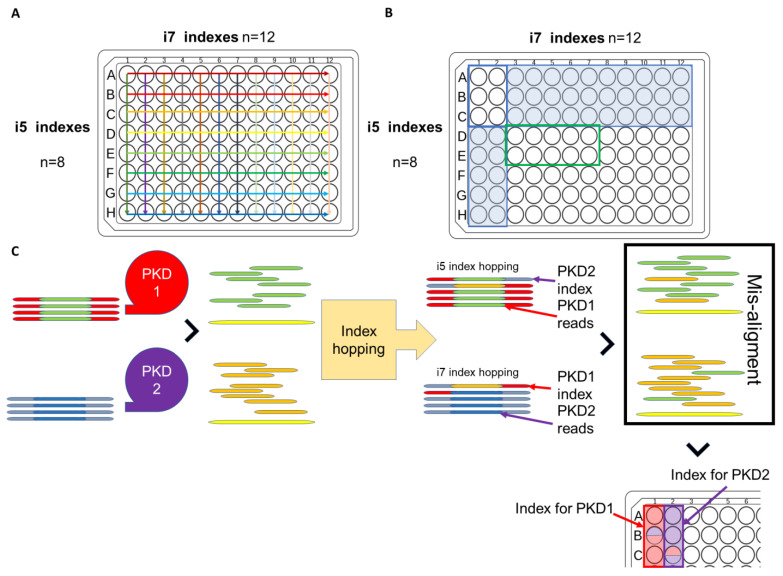
Combinatorial indexing and index hopping in a targeted sequencing panel [131]. (**A**) In a combinatorial index plate, i5 indexes are the same across the column, while the i7 index is the same across the row. (**B**) As a practice avoiding index carryover if A1 to C2 (6 wells in red rectangles) are adopted as indexes for a previous NGS run. Rows A to C, whose i5 indexes are the same (30 wells in blue), and columns 1 to 2, whose i7 indexes are the same (10 wells in blue), should be avoided. Indexes D3 to E6 (8 wells in a green rectangles) share a non-repetition of neither i5 nor i7 of the previous run and, hence, can be used for the current NGS run. (**C**) In the PKD1/PKD2-targeted sequencing panel, the differentiation of PKD1 and PKD2 of the same patient sample can be done by a 2-plex index, which PKD1 barcoded with well A1 and PKD2 barcoded with well A2. If index hopping occurs, the PKD1 read misligated with PKD2 index may be mapped as PKD2 is read, leading to the misalignment or false positive of a heterozygous variant.

**Table 1 cells-12-00493-t001:** Summary and comparison of some commercial kits and pros and cons of different sequencing techniques in COVID-19.

	Shotgun Metagenomics	Capture-Based Enrichment Targeted Sequencing	Amplicon-Based Enrichment Targeted Sequencing
Examples of commercial kits	Illumina Stranded Total RNA Prep with Ribo-Zero Plus	Illumina Respiratory Virus Oligo Panel Illumina Respiratory Pathogen ID/AMR Enrichment Panel Kit Roche KAPA SARS-CoV-2 Target Enrichment Panel Biosystem TWIST. SARS-CoV-2 Research Panel	Illumina COVIDSeq Test ThermoFisher Ion AmpliSeq™ SARS-CoV-2 Research Panel Paragon Genomics CleanPlex^®^ SARS-CoV-2 Panel Qiagen QIAseq SARS-CoV-2 Primer Panel
Characteristics			
Turnaround time	Long	Moderate	Short
Cost	High	Moderate to Low	Low
The complexity of the workflow	Moderate	Moderate to Low	Low
Coverage of the genome	High	Moderate with high uniformity	Low. with variable uniformity
Sequence depth	Low	High	High
The amount of starting material	High	Moderate to Low	Low
Sensitivity to the target	Low	High	High
Sensitivity to the background	High	Low	Low
Susceptibility to mutational effect	Low	High	High
Applications			
Track transmission	Yes	Yes	Yes
Identification of novel pathogen	Yes	No	No
Identification of co-infections and complex disease	Yes	Only Illumina respiratory panels	No
Identification of new mutations	Yes	Yes	No

**Table 2 cells-12-00493-t002:** Examples of species with identification issues using 16S rDNA sequencing [59].

Genus	Species
Aeromonas	*A. veronii*
Bacillus	*B. anthracis, B. cereus, B.globisporus, B. psychrophilus*
Bordetella	*B. bronchiseptica, B. parapertussis, B. pertussis*
Burkholderia	*B. cocovenenans, B. gladioli, B. pseudomallei, B. thailandensis*
Campylobacter	Non-jejuni-coli group
Edwardsiella	*E. tarda, E. hoshinae, E. ictaluri*
Enterobacter	*E. cloacae*
Neisseria	*N. cinerea, N. meningitidis*
Pseudomonas	*P. fluorescens, P. jessenii*
Streptococcus	*S. mitis, S. oralis, S. pneumoniae*

**Table 3 cells-12-00493-t003:** The pros and cons of using different TS approaches for a gene fusion analysis [79].

	Pros	Cons
Hybrid capture	Characterization of both known and unknown fusion variants of target genes Easily scalable to large gene panels Adequate for DNA and RNA gene fusion analysis At the DNA level, it does not require RNA purification and allows the simultaneous analyses of different gene variants	Higher RNA input than amplicon-based methods Difficulty with fusion variants involving large DNA intronic regions with repetitive sequences
Amplicon-based: Classical multiplex PCR (mPCR) Anchored multiplex OCR	Low RNA input Particularly effective with small and mid-size panels Analysis of both known and unknown fusion variants of target genes (anchored mPCR) 5′ and 3′ imbalance evaluation can increase test diagnostic accuracy	Not adequate for gene fusion analysis at the DNA level Primer design can be complex Characterization of only known fusion variants included in the panel (classical mPCR) PCR biases such as allele dropout can impact analysis results

## Data Availability

No new data were created or analyzed in this study. Data sharing is not applicable to this article.

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
