# Peer review of "Targeted Sequencing Approach and Its Clinical Applications for the Molecular Diagnosis of Human Diseases"

_cells, 2023, doi:10.3390/cells12030493_

Round 1

Reviewer 1 Report

The review related to Targeted sequencing (TS) and its clinical application is well designed and comprehensive. On the other hand, there are some concerns that should be addressed before the publication process.

1-     What year period was used to search the database to constitute the manuscript. There are some wide ranges of years in references like 1985 to 2022. Also, which keywords used to search literature.

2-     There is no information related to history and discovery of TS in the review.

3-     Application spectrum was seemed narrow for TS in the review. There is no example of TS application for other disease like neurodegenerative diseases, genetic diseases or other rare disease.

4-     Figure 3 missing.

5-     There could be more scheme to explain TS machinery or experimental procedures.  

Author Response

Dear Reviewer 1,

Please see the attachment for our answers to your comments.

Best Regards,

Martin

Reviewer 2 Report

Overall, the review is well-written and complete.

I would suggest some minor changes:

1)    It could be interesting for the reader to have some more specific information about the cost (e.g. cost per sample per gene; how the cost for multiple samples/genes is reduced;  how the cost of sequencing and analysis steps has decreased over the last years) and time needed to perform these TS approaches.

2) If possible, change the formatting of Table 1 to fit on one page.

Author Response

Dear Reviewer 2,

Please see the attachment for our answers to your comments.

Best Regards,

Martin

Round 2

Reviewer 1 Report

I have noting to add. The manuscript can be published in the present format.